# Associating Biological Activity and Predicted Structure of Antimicrobial Peptides from Amphibians and Insects

**DOI:** 10.3390/antibiotics11121710

**Published:** 2022-11-27

**Authors:** Amelia Richter, Darcy Sutherland, Hossein Ebrahimikondori, Alana Babcock, Nathan Louie, Chenkai Li, Lauren Coombe, Diana Lin, René L. Warren, Anat Yanai, Monica Kotkoff, Caren C. Helbing, Fraser Hof, Linda M. N. Hoang, Inanc Birol

**Affiliations:** 1Canada’s Michael Smith Genome Sciences Centre at BC Cancer, Vancouver, BC V5Z 4S6, Canada; 2British Columbia Centre for Disease Control, Public Health Laboratory, Vancouver, BC V6Z R4R, Canada; 3Department of Pathology and Laboratory Medicine, University of British Columbia, Vancouver, BC V6T 1Z4, Canada; 4Bioinformatics Graduate Program, University of British Columbia, Vancouver, BC V6T 1Z4, Canada; 5Department of Chemistry and the Centre for Advanced Materials and Related Technology, University of Victoria, Victoria, BC V8W 3V6, Canada; 6Department of Biochemistry and Microbiology, University of Victoria, Victoria, BC V8P 5C2, Canada; 7Department of Medical Genetics, University of British Columbia, Vancouver, BC V6T 1Z3, Canada

**Keywords:** antimicrobial peptide, AMP discovery, structure prediction, antimicrobial resistance

## Abstract

Antimicrobial peptides (AMPs) are a diverse class of short, often cationic biological molecules that present promising opportunities in the development of new therapeutics to combat antimicrobial resistance. Newly developed in silico methods offer the ability to rapidly discover numerous novel AMPs with a variety of physiochemical properties. Herein, using the rAMPage AMP discovery pipeline, we bioinformatically identified 51 AMP candidates from amphibia and insect RNA-seq data and present their in-depth characterization. The studied AMPs demonstrate activity against a panel of bacterial pathogens and have undetected or low toxicity to red blood cells and human cultured cells. Amino acid sequence analysis revealed that 30 of these bioactive peptides belong to either the Brevinin-1, Brevinin-2, Nigrocin-2, or Apidaecin AMP families. Prediction of three-dimensional structures using ColabFold indicated an association between peptides predicted to adopt a helical structure and broad-spectrum antibacterial activity against the Gram-negative and Gram-positive species tested in our panel. These findings highlight the utility of associating the diverse sequences of novel AMPs with their estimated peptide structures in categorizing AMPs and predicting their antimicrobial activity.

## 1. Introduction

Antimicrobial resistance is an escalating global health concern, with multiple infectious diseases becoming increasingly difficult and expensive to treat. An estimated 1.27 million deaths occurred in 2019 due to bacterial resistance to antibiotics [1], with the World Health Organization (WHO) stating that this number is expected to exceed 10 million by 2050 [2]. Antimicrobial resistance (AMR) develops through mutations in the bacterial genome as well as horizontal transfer of mobile genetic elements such as plasmids [3]. AMR is exacerbated by the routine administration of antibiotics both clinically and in agricultural settings [4]. Despite the steady increase in antibiotic resistant bacteria [5], there is a shortfall of novel antibiotics being developed, with no new classes approved for clinical use since the 1980s [6]. This gap between increasing antibiotic resistance and lagging discovery of new drug classes creates an urgent need for innovative therapeutic discovery methods to produce novel antimicrobials with different mechanisms of action to combat the emerging threat to public health [7].

Antimicrobial peptides (AMPs) are short, often cationic and amphipathic, biomolecules that are produced by the innate immune system of all living organisms [8]. They are a functionally and structurally diverse group of compounds that can defend against bacteria, viruses, fungi, and cancer [9]. AMPs can combat bacterial infections directly through interactions with the negatively charged membrane and with intracellular targets such as DNA and RNA [9,10]. In addition, AMPs can function indirectly through inflammatory or immunomodulatory pathways, resulting in the recruitment of immune cells [11]. Because of these diverse mechanisms of action, it has been suggested that it is more difficult for bacteria to develop resistance to AMPs as compared to conventional antibiotics [12,13]. However, bacteria intrinsically resistant to AMPs do exist [14], and resistance to colistin, an AMP-based therapeutic used as a last resort, has been observed [15]. Continuing the search for and characterization of AMPs with varying structure and associated mechanisms of action would increase our arsenal of available therapeutics against drug resistant bacteria.

The three-dimensional (3D) structures of AMPs can be classified into several types; α-helical, β-sheet containing, mixed, or linear extended structures [16]. It has been observed that many α-helical and linear extended peptides undergo conformational changes when interacting with the bacterial membrane [17]. The α-helix structure is reported to be the most effective conformation for AMPs to interact with the bacterial membrane, with size, sequence, charge and hydrophobicity all affecting the spectrum and level of the resulting antimicrobial activity [18]. However, the majority of discovered AMPs do not have a known structure, with approximately 25% of all entries in the Antimicrobial Peptide Database (APD3) reporting an experimentally determined structure [19]. Traditional methods for determining the structure of peptides, such as nuclear magnetic resonance (NMR), X-ray crystallography, and cryo-electron microscopy are laborious and expensive in comparison to in silico methods. While the latter only provide predictions, recent advances in artificial intelligence research bring the potential to make them enabling tools for characterizing AMPs. AlphaFold2 is a model that applies deep-learning to predict the 3D structure of proteins with high accuracy, including cases where there are no similar structures [20]. ColabFold has been reported to improve the speed of this prediction by coupling AlphaFold2 with an MMseqs2-based homology search [21].

Here, we studied the relationship between predicted AMP structure and observed antimicrobial activity. Specifically, we used ColabFold to predict the 3D structure of a list of 88 putative amphibian and insect AMPs discovered using rAMPage, a homology-based bioinformatic pipeline for in silico discovery of AMPs using RNA-seq reads [22]. We examined the antimicrobial activity of these peptides against two Gram-negative and one Gram-positive bacteria in the WHO’s list of priority pathogens [23]. We observe 51 AMPs in this list, discovered from 16 amphibian and 16 insect species, with antimicrobial activity and low toxicity to porcine erythrocytes and human embryonic kidney cells. Amphibians possess a diverse repertoire of AMPs, which provide a first line of defense as they transition from an aquatic to a terrestrial habitat throughout their life cycle [24]. Further, unlike vertebrates, insects lack an adaptive immune system, and as such their innate immune system, including AMPs, is their only line of defense in combating bacterial infections [25]. Studying these peptides, we report an association between AMPs with a predicted helical structure and broader antibacterial activity against the bacterial isolates of our panel.

## 2. Results

### 2.1. 3D Structure Prediction and Clustering

A total of 1137 putative AMPs were identified by rAMPage, and 88 peptides were selected for synthesis using three selection criteria: “Species Count”, “Insect Peptide”, or “AMPlify Score” (see Methods). The shortlist of 88 putative AMPs include 21 peptides described previously [22]. We investigated, in three independent experiments per peptide, the antimicrobial activity of the 88 peptides against *Escherichia coli* ATCC 25922, a clinical strain of *Salmonella enterica* serovar Enteritidis, and *Staphylococcus aureus* ATCC 29213. Fifty-one peptides showed antimicrobial activity against at least one of these bacterial strains (Table 1). To investigate the relationship between estimated structure and bioactivity, we predicted the 3D structures of the 88 mature peptides using ColabFold [21]. These were clustered based on their structural similarities evaluated by TM-score, and STRIDE [26] was used to assign the secondary structure of these predictions. The bioactivity of the peptides was classified as high when the median minimum inhibitory concentration (MIC) was ≤4 μg/mL, moderate when 4 < median MIC ≤ 16 μg/mL, and low when 16 < median MIC ≤ 128 μg/mL. As the highest concentration of peptide tested was 128 μg/mL, a median MIC of >128 μg/mL was classified as ‘no inhibition observed’ (N/O). Approximately half (45/88) of the peptides were predicted to adopt a helical structure, three adopted a structure containing β-strands, 17 adopted a linear extended structure, and 23 peptides were predicted to contain both helical and linear extended regions (Figure 1). The latter 23 peptides were classified as either mainly extended (>50% residues fell in extended category; six peptides) or mainly helical (between 50–80% residues in helical structures; 17 peptides). All structural categories contained both amphibian- and insect-derived peptides of varying length, charge, and hydrophobicity profiles. There was an association between antimicrobial activity and structure, with 8% of active peptides being linear extended and 71% being helical, despite extended peptides accounting for 19% of total peptides and helical only accounting for 51%. This association was statistically significant at the alpha = 0.05 level, with a *p*-value = 6.6 × 10^−5^. Predicted helical/extended content did not correlate with observed in vitro antimicrobial activity when the percentage of helical residues was less than 80%. Approximately half of the mainly helical and mainly extended peptides were active.

### 2.2. Antimicrobial Susceptibility Testing and Cytotoxicity

We tested the 88 putative AMPs for their antimicrobial activity using a broth microdilution assay [27,28]. The panel of bacteria tested included two Gram-negative strains: *E. coli* ATCC 25922 and *Salmonella* Enteritidis. Additionally, we tested one Gram-positive strain: *S. aureus* ATCC 29213. Of the 88 peptides tested, 51 displayed antimicrobial activity (MIC ≤ 128 µg/mL) against at least one of the pathogens, including 35 from amphibian and 16 from insect sources (Figure 2A). All AMPs that only inhibited growth of *E. coli* (10 peptides) had low activity. Eleven peptides were selective against the Gram-negative bacteria, with no inhibitory activity against *S. aureus* at the tested concentrations. Further, 30 peptides were active against both Gram-negative and Gram-positive species, with 5 having no inhibitory effect on *Salmonella* Enteritidis and 25 being active against all three species. Bioactivity quantitative data is displayed in Appendix A. Additionally, the hemolytic 50 concentration (HC50, see Methods for definition) of the peptides was determined with porcine red blood cells as a cost-effective initial toxicity assessment of the AMPs. Six of the active peptides displayed low hemolytic activity, with the HC50s ranging from 64–128 µg/mL.

A total of 14 peptides displayed high antimicrobial activity against at least one of the pathogens tested. The three most active amphibian peptides, OdMa2, PeNi1 and PeNi9, were broadly active. OdMa2 and PeNi1 had MICs of 4 µg/mL against *E. coli* and 8–16 µg/mL against *Salmonella* Enteritidis. OdMa2 had an MIC of 4–8 µg/mL and PeNi1 had an MIC of 4 µg/mL against *S. aureus*. PeNi9 had MICs of 2–4 µg/mL against *E. coli*, 8–16 µg/mL against *Salmonella* Enteritidis, and 4–8 µg/mL against *S. aureus*. OdMa2 was slightly hemolytic, having an HC50 = 128 µg/mL, while PeNi1 and PeNi9 did not show hemolytic activity at the highest concentration tested. The three most active insect peptides were PaVa1, TeBi1 and TeRu4. PaVa1 and TeRu4 were selectively active against Gram-negatives, with MICs of 2–4 µg/mL and 1 µg/mL against *E. coli* and 4–8 µg/mL and 2–4 µg/mL against *Salmonella* Enteritidis, respectively. These two peptides did not inhibit growth of *S. aureus* at the concentrations tested. TeBi1 was broadly active, with an MIC of 1–2 µg/mL against *E. coli*, 4–8 µg/mL against *Salmonella* Enteritidis and *S. aureus*. All top three insect peptides were not hemolytic at the concentrations tested.

The six most active peptides were tested for cytotoxicity against the human embryonic kidney cell line HEK293 using the alamarBlue cell viability assay. PeNi1 and OdMa2 were toxic to less than 10% of cells up until 32 µg/mL, where cell viability decreased to <75% at 64 µg/mL and to approximately 0% at 128 µg/mL (Figure 2B). More variation in cell viability was observed at 64 µg/mL for these peptides compared to other concentrations. Cell viability when exposed to PeNi9 and TeBi1 was, on average, 0% and 40% at 128 µg/mL, respectively. These peptides had low or no cytotoxicity at lower concentrations. We observed that TeRu4 and PaVa1 were not toxic to HEK293 cells at any of the concentrations tested.

### 2.3. Sequence and Structural Characterization of AMPs Discovered Using rAMPage

To investigate the sequence diversity of the 51 peptides displaying antimicrobial activity, we performed a multiple sequence alignment of the mature amino acid sequences and generated a phylogenetic tree. Peptides derived from amphibian/insect datasets mostly clustered with other peptides from the same datasets (Figure 3A). To investigate whether the mature sequences were similar to any known proteins, we performed a local BLASTp [29] search of the validated AMPs. Approximately two thirds of the peptides were novel, with less than 100% sequence identity to their top BLAST hits (Figure 3A). Of these peptides, six amphibian-derived and seven insect-derived AMPs had no significant hits in the non-redundant protein database. CaCa1 and CaCa2 had 100% sequence identity with their top BLAST hits but no antimicrobial activity was indicated in the NCBI entries. OdMa13, PeNi11, OdMa5, OdMa6, OdMa7 and RaCa15 were identical to AMPs seen in other organisms of the same genera (Appendix A). OdMa10, an AMP discovered in the frog *Odorrana margaretae*, was identical to the AMP nigrosin-MG1 from this species. PeNi8, PeNi10, and PeNi7, derived from the black-spotted frog *Pelophylax nigromaculatus*, aligned with 100% sequence identity to the mature region of the AMPs pelophylaxin-2, ranatuerin-2N, and nigrocin-6N, respectively, from this species. Of note, we also discovered the mature sequence of PeNi8 in five other amphibian species, PeNi10 in four other amphibian species, and PeNi7 in six other amphibian species. TeBi1, PeNi3 and RaOm3 also had BLAST hits with 100% sequence identity to precursors of known antimicrobial peptides, however they were not identical to the mature bioactive peptide. TeBi1 included the sequence of the mature AMP bicarinalin from the same species, but at its C-terminus it also contained three amino acids present in the peptide precursor but not in the mature sequence. Of their top BLAST hit esculentin-2N, a 36-residue AMP, PeNi3 and RaOm3 only spanned 18 and 19 residues, respectively. Additionally, PaVa1 had 100% sequence identity to a region of an apidaecin type 14-like isoform; however no mature region was annotated in this result and no in vitro validation of antimicrobial activity was available from public repositories.

Further, we analyzed our peptide sequences with InterProScan to identify protein family domains and classifications. We discovered 18 active amphibian peptides containing the signature for the *Ranidae* AMP Nigrocin-2 family, three from the Brevinin-1 family, and seven with the Brevinin-2 signature. Additionally, we classified two insect peptides as being a part the Apidaecin family. While PaVa1 was not identified as containing any family signatures by InterProScan, we classified it as an Apidaecin as it possessed the RP… PRPPHPRL motif that has been identified as being conserved in Apidaecins [30]. Most Nigrocin-2-like peptides (15/18) were 21 amino acids long (Figure 3B) with all 21 residues identified as part of the family signature. OdMa4, which is 14 amino acids long, aligns with the N-terminal of the other 15 peptides and 100% of its length is identified as containing the signature. In contrast, the two peptides longer than 21 amino acids, OdTo4 and PeNi5, only have part of their sequences containing the signature. The peptides characterized as belonging to the Brevinin-1 family were longer and contained more hydrophobic residues than most of the Nigrocin-2 peptides within our set (Figure 3C). All peptides classified as being part of the Brevinin-2 family (Figure 3D) were longer than those in the Nigrocin-2 and Brevinin-1 families, containing 29–30 amino acids and 1–2 negatively charged residues.

To explore the relationship between predicted helical or extended structure and activity, we investigated the distribution of bioactive peptides for each bacterial species tested in relation to the percentage of residues contained in helical or extended structures. Peptides with higher extended content were active against *E. coli* and *Salmonella* Enteritidis, but not *S. aureus* (Figure 4). All peptides with activity against *S. aureus* were predicted to have some helical secondary structure, with the majority having >70% of the peptide assigned as having helical structure. Peptides identified as containing the signatures for the Apidaecin, Nigrocin-2, Brevinin-1 and Brevinin-2 families appeared to possess similar bioactivity profiles and predicted secondary structure content to other members within their families. Superimposed predicted structures for the family members can be found in Appendix A. The two Apidaecin peptides were linear extended and selective to the Gram-negative bacteria tested, with moderate to high activity against both *E. coli* and *Salmonella* Enteritidis. The Brevinin-2 and Nigrocin-2 peptides were mostly helical and broadly active against all three bacteria with the exception of OdMa4, which is mainly extended and displayed selective antimicrobial activity against the Gram-negative bacteria tested. All of the Brevinin-1 peptides were mostly helical and active against *E. coli*, non-inhibitory against *Salmonella* Enteritidis, and two out of three were active against *S. aureus*.

## 3. Discussion

In the present study, we characterized 51 AMP candidates originally derived from a bioinformatics scan of RNA-seq data from amphibians and insects using rAMPage [22]. Fourteen of these had high antimicrobial activity to at least one of the bacterial species tested. The most bioactive amphibian AMPs, PeNi1, OdMa2 and PeNi9, and the insect peptide TeBi1 were active against all three bacteria. TeRu4 and PaVa1 both demonstrated selectivity for the Gram-negative species tested, with TeRu4 having higher antimicrobial activity against both strains.

The AMPs reported herein had low to no toxicity, with only six peptides having hemolytic activity at the highest concentrations tested. This observation is consistent with previous research that suggests that AMPs tend to have selectivity for microbial cells over eukaryotic cells due to differences in membrane composition [31]. Despite this selectivity, some AMPs do disrupt mammalian membranes and cell processes [32]. In the reported list of AMPs, PeNi1, OdMa2, PeNi9 and TeBi1 only displayed high levels of cytotoxicity at concentrations higher than their MICs. The ratio of the toxicity to the MIC of the peptides is described using the selectivity index, with a larger selectivity index indicating preference for bacterial cells [33,34] (Appendix A). TeBi1 displayed the most selectivity of these, as it was not toxic to over 50% of cells until 16 to 128-fold its MIC. TeRu4 and PaVa1 were not toxic against either porcine erythrocytes or HEK293 cells. The selectivity indices of TeBi1, TeRu4 and PaVa1 make them good candidates for therapeutic development.

The AMPs characterized here vary in their amino acid sequences, with 30 active peptides belonging to four AMP families. AMPs from the *Ranidae* family can be classified into 14 peptide families based on their amino acid sequence, with AMPs from the same peptide family thought to share a common evolutionary origin [35]. We identified AMPs from three of these families: Brevinin-1, Brevinin-2 and Nigrocin-2 using the families’ signatures. Brevinin-2 peptides are the longest of the three, with a mature peptide length of 33–34 residues [36]. Brevinin-1 and Nigrocin-2 peptides are shorter, with an approximate length of 24 and 21 amino acids, respectively [36]. However, there is considerable variation among individual family members [36]. AMPs from these three families share some common characteristics: presence of a C-terminal disulfide-bridged cyclic heptapeptide, otherwise known as the ‘Rana box’, and antimicrobial activity against both Gram-negative and Gram-positive bacteria [36,37]. Additionally, Brevinin-1, Brevinin-2 and Nigrocin-2 peptides have been found to adopt an amphipathic α-helical conformation in membrane-like environments [36,37,38]. All peptides we identified as members of these families included the Rana box and were predicted to have a helical or mainly helical structure except for the Nigrocin-2 peptide OdMa4. OdMa4 was the shortest of the Nigrocin-2 peptides and terminated after the first cysteine of the Rana box. Additionally, we identified two peptides belonging to the Apidaecin AMP family. Apidaecins are short, proline-rich AMPs produced by insects [30]. These peptides do not typically form α-helices or β-strands [30], as was seen with ApCe1 and PaVa1, which were predicted to have linear extended structures. Similar to other Apidaecins, both ApCe1 and PaVa1 showed antimicrobial activity against Gram-negative bacterial species but not Gram-positives [30].

Most of the validated AMPs were predicted to adopt a helical structure by ColabFold, despite helical structures only making up half of the peptide set. In contrast, predicted linear extended peptides made up a smaller proportion of the antimicrobially active peptides compared to their abundance in the overall set. Note that, while there was a tendency of peptides with a predicted helical structure to be bioactive, not all of the helical peptides were active, and vice versa. In addition, we observed that while there were peptides from all structural categories that were active against Gram-negative species, only peptides that had some predicted helical content were active against both Gram-positive and Gram-negative bacteria. AMPs that form α-helices make up the majority of AMPs with known structure, and it is thought to be the most effective structure rendering antibacterial activity against the bacterial membrane [18]. However, this also biases machine learning algorithms such as the one we used to discover the tested peptides in favour of peptides with α-helix structures.

The majority of known AMPs do not have an experimentally validated structure [19]. Determination of peptide structures is often accomplished with NMR, X-ray crystallography or cryo-EM; however, this is time consuming and costly in comparison to in silico techniques. In contrast, one can predict the secondary and tertiary structures in a high-throughput manner based on the amino acid sequences of peptides of interest using a tool like ColabFold [21] prior to synthesis and testing. The association between ColabFold predicted structures and antimicrobial activity identified here can be informative in selecting peptides identified in silico as putative AMPs for in vitro validation. Further, AMPs are often chemically modified to improve their stability and bioavailability. The insights into the predicted structure-function relationships identified here could also be used to investigate how chemical modifications may influence antimicrobial activity.

One of the limitations of using rAMPage for peptide discovery is that the pipeline uses RNA-seq reads, and thus does not detect post-translational modifications such as amidation. Amidation of the C-terminus is a common post-translational modification of AMPs [39], and can impact the antimicrobial activity and toxicity of peptides [40]. PeNi12 for example, has 100% sequence identity to the mature region of ranacyclin-N from *P. nigromaculatus*, however PeNi12 was non-inhibitory against our panel of bacteria at all concentrations tested. Ranacyclins have amidated C-termini [41], so the non-amidated carboxyl end may have contributed to the lack of antimicrobial activity of PeNi12. TeBi1 has 100% sequence identity to the C-terminus region of the propeptide of bicarinalin and includes the mature sequence of bicarinalin, however the last three amino acids at the C-terminus of bicarinalin precursor are cleaved off and the peptide is amidated [42]. Additionally, as the peptides were only tested *in vitro,* we cannot conclude that the peptides that did not exhibit antimicrobial activity are not AMPs as they may have immunomodulatory properties or act on other bacterial strains not tested in the present study. The AMPs identified here may be tested in vivo to determine if they interact with the immune system to combat infections within the host.

While ColabFold determines protein structures in silico with high accuracy, peptides in their biological environments are flexible and may adopt different conformations [17]. AMPs that adopt a helical structure at the membrane are often disordered in aqueous environments [17]. Thus, the predicted structure may not be representative of the 3D structure of the peptide when interacting with specific targets. Additionally, AMPs have diverse sequences and targets. Peptides that act on the bacterial membrane may have different structural characteristics compared to those that mainly interact with intracellular components. Future investigations into the mechanism of action of these AMPs would shed light into how the predicted secondary structure impacts their biological function. The bioinformatic characterization of these AMPs is a cost-effective initial step in studying their mechanisms of action.

AMPs are a promising alternative to conventional small molecule antibiotics as they represent a diverse group of molecules with a wide variety of physiochemical properties. Here, we present the discovery and characterization of structurally and functionally diverse AMPs with potent antimicrobial activity and low toxicity. We also demonstrate the utility of predicting the structure of AMPs and report a significant association between peptides predicted to adopt a helical conformation and observed antimicrobial activity (*p* = 6.6 × 10^−5^). The potential of structural prediction to prioritize putative AMPs is an exciting avenue for discovery of new therapeutics in our fight against bacterial infections.

## 4. Materials and Methods

### 4.1. Peptide Discovery Using rAMPage

Peptide sequences were discovered using the Rapid Antimicrobial Peptide Annotation and Gene Estimation (rAMPage) pipeline v1.0 [22], which is publicly available on Github (https://github.com/bcgsc/rAMPage, accessed on 14 February 2021). rAMPage is a homology-based pipeline that uses RNA-seq reads as input to generate putative AMPs for downstream in vitro testing. Briefly, rAMPage generates putative AMPs by first processing the input RNA-seq reads using fastp v0.20.0 [43] and assembles the reads into transcripts with RNA-bloom v1.3.1 [44]. Transcripts are subsequently translated by Transdecoder v5.5.0 [45] and a homology search is conducted with HMMER v3.3.1 [46]. Precursor sequences are cleaved with ProP v1.0c [47] and putative mature AMP sequences are prioritized by AMPlify v1.0.3, a deep-learning classifier [48].

The putative AMPs were filtered to retain peptides with a minimum charge of +2 and maximum length of 30 amino acids. Peptides were further characterized with ENTAP v0.10.7 [49], Exonerate v2.4.0 [50] and SABLE v4.0 [51] before being clustered with CD-HIT v4.8.1 [52]. Ninety sequences were prioritized for synthesis from this list using three selection criteria:“Species Count”—peptides identified in two or more species;“Insect Peptide,”—insect-derived peptides chosen using a reduced AMPlify prediction score threshold; and“AMPlify Score”—the top-scoring peptides with the highest net positive charge.

Peptides were purchased from GenScript and provided in lyophilized 0.08 milligram aliquots. Two peptides were unable to be synthesized, resulting in a final experimental set of 88 putative AMPs.

### 4.2. Bacterial Isolates

Bacteria were grown overnight at 37 °C in Mueller-Hinton Broth (MHB; Sigma-Aldrich, St. Louis. MO, USA) in a shaking incubator and aliquoted into cryovials with a 50% glycerol solution in a 1:1 ratio. No additives were added to the MHB during the growth of bacterial isolates. Glycerol stocks were stored at −80˚C.

### 4.3. Antimicrobial Susceptibility Testing (AST)

The antimicrobial activity of the putative AMPs was determined by a broth microdilution assay as described by the Clinical and Laboratory Standards Institute [27], with the published adaptations for testing of cationic peptides [28]. *Escherichia coli* 25922 and *Staphylococcus aureus* 29213 isolates were purchased from the American Type Culture Collection (ATCC; Manassas, VA, USA). A human clinical isolate of *Salmonella enterica* serovar Enteritidis was provided by the BC Centre for Disease Control. Bacteria from stocks stored at −80 °C were streaked onto nonselective Columbia blood agar with 5% sheep blood (Oxoid) and incubated overnight at 37 °C. Next, 2–4 colonies were streaked onto a new agar plate to ensure uniform health of colonies used in the broth microdilution assay. Isolated colonies were suspended in MHB and the optical density was measured with a spectrophotometer to create an initial inoculum concentration of approximately 1 *×* 10^8^ cfu/mL. The inoculum was diluted 1:250 to a final concentration of 2–8 × 10^5^ cfu/mL, which was confirmed by performing a Total Viability Count (TVC). The TVC consisted of plating a 1:1000 dilution of the final inoculum on nonselective media and was also used to confirm the health and purity of the inoculum in each trial.

Lyophilized 80 µg peptide aliquots synthesized by GenScript (Piscataway, NJ, USA) were resuspended to 1.28 mg/mL with UltraPure water (Thermo Fisher Scientific, Waltham, MA, USA) and serially diluted in polypropylene 96-well microtiter plates (Greiner Bio-One #650261, Kremsmünster, Austria) from 128 down to 0.5 µg/mL. Two columns per plate were reserved for growth and sterility controls. Inoculum was added to the wells containing the peptide and the growth control. Plates were incubated for 20–24 h at 37 °C. The minimum inhibitory concentration (MIC) was determined as the lowest peptide concentration where there was no visible bacterial growth.

### 4.4. Structure Prediction and Clustering

We predicted the 3D structures of the mature peptides using a local installation of ColabFold [21]. Using ColabFold’s server, five structures were generated for each peptide with sub-models of AlphaFold. Structural templates were used as input along with the peptide sequences for better accuracy. The five estimated structures were relaxed using the Amber force field, which helps remove stereochemical violations in predictions [20]. The five structures for each sequence were ranked by the per-residue estimate of AlphaFold’s confidence in its prediction (pLDDT score). The PDB file of the Amber relaxed rank 1 model for each peptide was used for further analysis.

Similarity of the predicted structures of peptides was determined using mTM-align (version 20180725) [53], which was used to prepare a distance matrix calculated from the pairwise TM-scores of the PDB files produced for each peptide. This distance matrix was clustered using scipy.cluster.hierarchy [54] in python with complete linkage to produce a dendrogram. PDB files were fed into STRIDE [26] and the residue assignments were used to categorize peptides. Peptides were classified into linear extended peptides (only possessing turn secondary structure and/or coils), mainly extended (>50% of the residues were in the extended category), mainly helical (between 50–80% of the residues were helical), helical (>80% of the residues were assigned as participating in an α-, pi- or 310-helical structure), or β-strand containing. The fisher.test() function in base R was used to conduct a Fisher’s exact test to investigate the independence of the helical or linear extended structure and possession of activity. Only peptides classified as helical (≥80% of residues assigned helical, 45 peptides) or linear extended (no residues in helical or β-strand structures, 17 peptides) were included in this statistical test. Peptides were labeled as active if they visually inhibited at least one of the bacterial strains at one of the tested concentrations.

### 4.5. Hemolysis Assay

Peptides were evaluated for toxicity using three independent hemolysis experiments. Whole blood from healthy donor pigs, supplemented with Na Citrate, was purchased from Lampire Biological Laboratories (Pipersville, PA, USA). Red blood cells (RBCs) were washed and isolated by three centrifugation cycles using Roswell Park Memorial Institute medium (RPMI; Thermo-Fisher Scientific) to create a 1% (*v*/*v*) RBC solution. Lyophilized AMPs were suspended and serially diluted from 128 down to 1 μg/mL using RPMI in a 96-well plate, before being combined with 100 μL of the 1% RBC solution. Following a minimum 30 min incubation at 37 °C, plates were centrifuged and ½ volume from each supernatant was transferred to a new 96-well plate. The absorbance of these wells was measured at 415 nm. To quantify hemolytic activity and determine the AMP concentration that lyses 50% of the RBCs (HC50), absorbance reading from wells containing RBCs treated with 11 μL of a 2% TritonX-100 detergent solution (TX-100) or RPMI (AMP solvent-only) were used to define 100% and 0% hemolysis, respectively. All centrifugation steps were performed at 500× *g* for five minutes in an Allegra-6R centrifuge (Beckman Coulter, Brea, CA, USA).

### 4.6. Cytotoxicity

The human embryonic kidney cell line HEK293 and the corresponding growth media and supplements were purchased from the ATCC. The cells were maintained in Eagle’s Minimum Essential Medium (EMEM) supplemented with 10% fetal bovine serum (FBS) and 1% Penicillin-Streptomycin solution and grown in a 5% CO_2_ incubator at 37 °C. Approximately 1 × 10^4^ cells were distributed to each well of a 96-well flat-bottom cell culture plate (Corning #3595, Corning, NY, USA). Cells were incubated overnight to allow them to adhere. Lyophilized 80 µg peptide aliquots were resuspended to 1.28 mg/mL with UltraPure deionized water and serially diluted in polypropylene 96-well microtiter plates from 128-0.5 µg/mL. TX-100 was diluted to 2% in UltraPure deionized water and added as a positive control. Complete growth medium was added to the peptides and TX-100 containing wells. Spent growth medium of wells with adhered cells was replaced with the contents of wells containing serially diluted peptides or TX-100. Cells were incubated for four hours at 37 °C in 5% CO_2_ incubator. Growth medium containing peptides or TX-100 was replaced with fresh growth medium containing 10% (*v*/*v*) alamarBlue (Bio-Rad Laboratories, Hercules, CA, USA) and incubated for 20 h. Fluorescence was measured at excitation of 540 nm and emission of 590 nm in a Cytation 5 plate reader (Agilent BioTek, Winooski, VT, USA) and results recorded and analyzed with Gen5 software (Agilent BioTek, Winooski, VT, USA). The average fluorescence reads of TX-100 and growth media only wells were used to calculate 0% and 100% cell viability, respectively. The percentage of viable cells at each peptide concentration was determined by:$$\mathrm{cell}\;\mathrm{viability}\left(\%\right)=\left(1-\frac{{\mathrm{fluorescence}}_{\left(100\%\;\mathrm{viable}\right)}-{\;\mathrm{fluorescence}}_{\left(\mathrm{peptide}\right)}}{{\mathrm{fluorescence}}_{\left(100\%\;\mathrm{viable}\right)}-{\mathrm{fluorescence}}_{\left(0\%\;\mathrm{viable}\right)}}\right)\times 100\%$$

### 4.7. BLAST and Phylogenetic Analysis

Amino acid sequences for the mature peptides tested in vitro were used as the query sequences for BLASTp analysis with NCBI BLAST+ v2.13.0 [29]. The sequences were searched against the non-redundant protein database (downloaded on 01/19/2022) [29]. Max target sequences was set to 5. To identify protein family memberships, mature sequences were fed into InterProScan v5.56-89.0 [55] in FASTA format with default parameters. Multiple sequence alignment was performed with the 51 validated AMPs using the Bioconductor R package msa v1.28.0 [56] with the ClustalW option. A distance matrix was created using the multiple sequence alignment with the seqinr package v5.6-2 [57] and a phylogenetic tree using neighbour-joining tree estimation was created using the ggtree v3.4.2 [58,59,60,61] and ape v5.6-2 [62] R packages. AMP protein families, origin of AMPs and BLASTp results were annotated on the phylogenetic tree using ggtree [58,59,60,61].

## Figures and Tables

**Figure 1 antibiotics-11-01710-f001:**
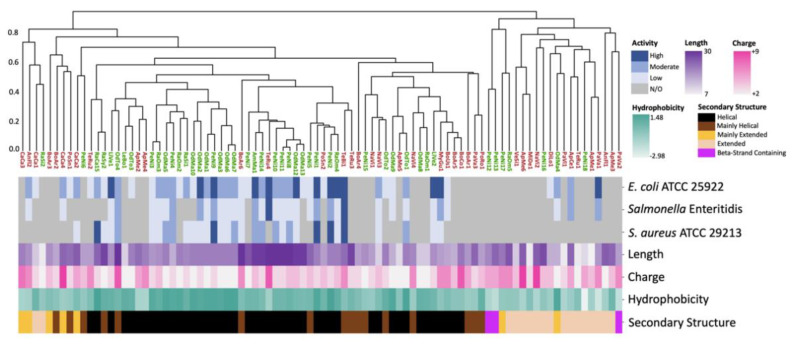
Agglomerative hierarchical clustering of peptides based on their predicted 3D structures. The bioactivity of each peptide against the three organisms tested (*E. coli* ATCC 25922, *Salmonella* Enteritidis, and *S. aureus* ATCC 29213) is reported in the top three rows. Bioactivity is assigned based on the median MIC and categorized as high for peptides with a MIC ≤ 4 µg/mL, moderate for 4 < MIC ≤ 16 µg/mL and low for 16 < MIC ≤ 128 µg/mL. Peptides that did not inhibit bacterial growth at tested concentrations were classified as ‘no inhibition observed’ (N/O). Physiochemical attributes of the AMPs tested (length, charge, and hydrophobicity) are displayed in rows 4, 5 and 6 using purple, pink and green gradients, respectively. The colour-coded peptide secondary structures (helical, mainly helical, mainly extended, linear extended and β-strand containing) shown in row 7 were assigned based on 3D coordinates of predicted structures. AMP name labels below the dendogram are colour-coded to indicate their amphibian (green) or insect (brown) origin.

**Figure 2 antibiotics-11-01710-f002:**
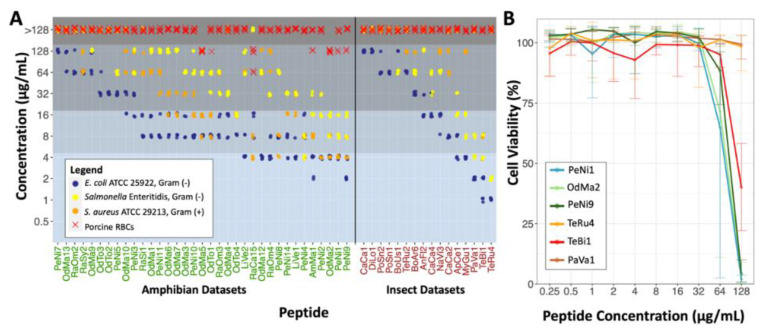
Antimicrobial activity and mammalian cell toxicity of peptides discovered. (**A**) Minimum inhibitory and hemolytic 50 concentration (MIC and HC50, respectively) of peptides with antimicrobial activity against at least one bacterial strain within a panel composed of *E. coli* ATCC 25922, *Salmonella* Enteritidis and *S. aureus* ATCC 29213. HC50 was determined using porcine red blood cells (RBCs), as described in the Methods. Bioactive peptides identified from amphibian (green text) and insect (brown text) datasets are shown. Peptide bioactivity levels are separated by grey shaded blocks. From top to bottom, the dark grey block indicates no observable activity, the medium-dark grey block includes low activity, the medium grey block designates moderate activity, and the light grey block corresponds to high antimicrobial activity. (**B**) Mean relative cellular viability of HEK293 (cultured human kidney cells) after 24 h incubation with the six most active insect and amphibian peptides at concentrations up to 128 µg/mL from three independent experiments. Error bars indicate range of data.

**Figure 3 antibiotics-11-01710-f003:**
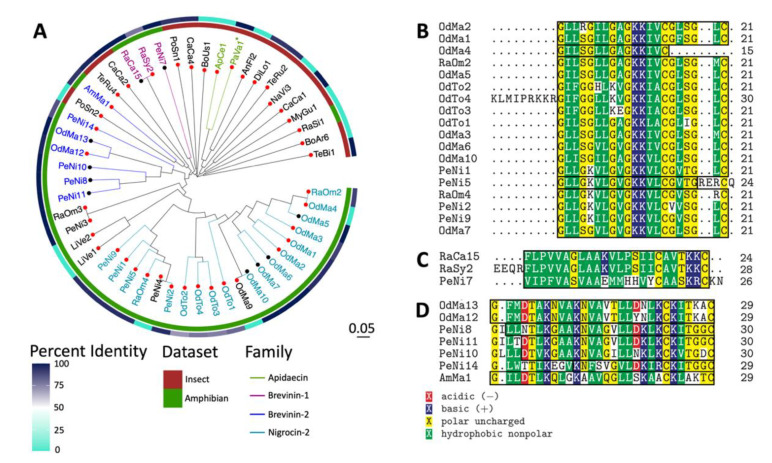
Phylogenetic analysis of validated AMPs. (**A**) Circular phylogenetic tree based on a multiple sequence alignment of active rAMPage peptides with family classifications highlighted (colour-coded AMP names). The origin of AMP (insects in brown and amphibian in green) and BLASTp percent identity to entries in the NCBI non-redundant protein database (on a scale of 0–100%) are indicated by the inner and outer rings, respectively. Tip points of peptides that do not have 100% sequence identity to the bioactive region of known AMPs are highlighted in red. Multiple sequence alignments of the Nigrocin-2 (**B**), Brevinin-1 (**C**) and Brevinin-2 families (**D**), with the outlines indicating the identified family signature.

**Figure 4 antibiotics-11-01710-f004:**
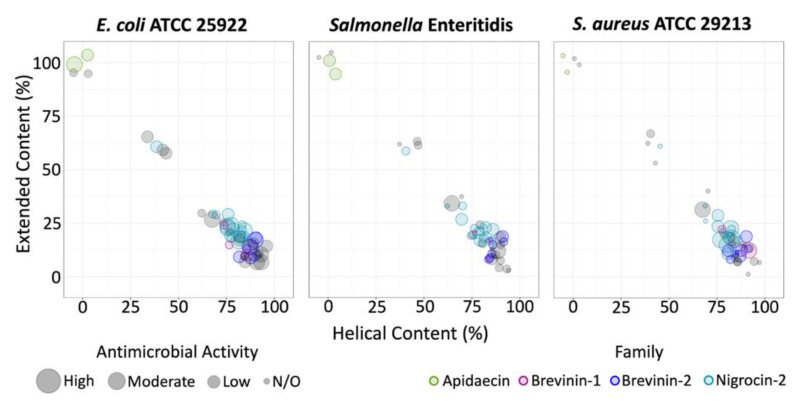
Predicted structural content and antimicrobial activity of 51 bioactive peptides against *E. coli*, *Salmonella* Enteritidis, and *S. aureus*. Secondary structure content indicated by the percentage of residues assigned to participate in helical vs. linear extended regions. Distribution of AMP family members and level of antimicrobial activity are distinguished by colour and size of circles, respectively.

**Table 1 antibiotics-11-01710-t001:** Sequences and physiochemical properties of the 51 bioactive AMPs discovered using rAMPage.

Peptide Name	Source Organism	Sequence	Length	Charge *	AMPlify Score	MW (Da)
AmMa1	*Amolops mantzorum*	GILDTLKQLGKAAVQGLLSKAACKLAKTC	29	4	80.0	2943.59
AnFl2	*Anterhynchium flavomarginatum*	GILRSLGWIQMPRSRRRHR	19	6	31.8	2375.82
ApCe1	*Apis cerana*	GIYTGRLLPVYIPQPRPPHPRLRR	24	5	38.6	2853.39
BoAr6	*Bombus ardens*	GILRLVTRRFRFSPTNLNRYTVARLVSGVP	30	6	22.1	3460.06
BoUs1	*Bombus ussurensis*	RKIIAVSVHKLCRVKR	16	6	29.9	1906.40
CaCa1	*Camponotus castaneus*, *Odontomachus monticola*, *Polistes rothneyi, Polistes snelleni, Sphecidae *sp. KJ-8906, *Vespa dybowskii*	FACPIGFFRLKR	12	3	7.27	1454.79
CaCa2	*Camponotus castaneus*, *Odontomachus monticola*, *Temnothorax rugatulus*	FIKTQVLKHLVAGVRVARGLDWKWR	25	5	28.7	2977.57
CaCa4	*Camponotus castaneus*	RRFFFATAPCGYSRKFCKITRRKR	24	9	23.6	2996.58
DiLo	*Diachasmimorpha longicaudata*	GAFVLWGPTPRPRRR	15	4	26.0	1766.07
LiVe1	*Litoria verreauxii*	GWLDIAKKVASVVAGIVKR	19	3	80.0	2010.44
LiVe2	*Litoria verreauxii*	GWLDIAKKVASVVAGLGKR	19	3	70.0	1968.36
MyGu1	*Myrmecia gulosa*	RRAIFASIRGYLGLRKR	17	6	25.3	2033.44
NaVi3	*Nasonia vitripennis x Nasonia giraulti* F1	KLFLTLWKLKR	11	4	30.5	1445.84
OdMa1	*Odorrana margaretae*	GLLSGILGAGKKIVCGFSGLC	21	2	80.0	1993.45
OdMa2	*Odorrana margaretae*	GLLRGILGAGKKIVCGLSGLC	21	3	67.0	2028.54
OdMa3	*Odorrana margaretae*	GLLSGLLGAGKKIVCGLSGMC	21	2	80.0	1977.47
OdMa4	*Odorrana margaretae*	GILSGLLGAGKKIVC	15	2	70.0	1428.79
OdMa5	*Odorrana margaretae*	GILSGLLGAGKKIVCGLSGLC	21	2	80.0	1959.43
OdMa6	*Odorrana margaretae*	GLLSGVLGVGKKIVCGLSGLC	21	2	80.0	1973.46
OdMa7	*Odorrana margaretae*	GLLSGVLGVGKKVLCGLSGLC	21	2	80.0	1973.46
OdMa9	*Odorrana margaretae*	GLISGILGAGKKVLC	15	2	67.0	1428.79
OdMa10	*Odorrana margaretae*	GLISGILGAGKKVLCGLSGLC	21	2	70.0	1959.43
OdMa12	*Odorrana margaretae*	GFMDTAKNVAKNVAVTLLYNLKCKITKAC	29	4	70.0	3158.82
OdMa13	*Odorrana margaretae*	GFMDTAKNVAKNVAVTLLDNLKCKITKAC	29	3	67.0	3110.73
OdTo1	*Odorrana tormota*	GILSGLLGAGKKLACGLIGLC	21	2	80.0	1957.46
OdTo2	*Odorrana tormota*	GIFGGHLKVGKKIACGLSGLC	21	3	67.0	2058.52
OdTo3	*Odorrana tormota*	GIFGGLLKEGKKIACGLSGLC	21	2	48.5	2064.53
OdTo4	*Odorrana tormota*	KLMIPRKKRGIFGGLLKVGKKIACGLSGLC	30	8	47.4	3186.06
PaVa1	*Partula varia*	RPRPQQVPPRPPHPRLRR	18	6	27.5	2240.63
PeNi1	*Pelophylax nigromaculatus*	GLLGKVLGVGKKVLCGVTGLC	21	3	70.0	2014.55
PeNi2	*Pelophylax nigromaculatus*	GLLGKVLGVGKKVLCVVSGLC	21	3	70.0	2042.61
PeNi3	*Pelophylax nigromaculatus*	GIFSLIKGAAKVVAKGLG	18	3	65.2	1729.12
PeNi4	*Pelophylax nigromaculatus*	GLLGKVLGVGKKVLC	15	3	67.0	1483.91
PeNi5	*Pelophylax nigromaculatus*	GLLGKVLGVGKKVLCGVTGRERCQ	24	4	57.7	2471.01
PeNi7	*Bufo gargarizans*, *Leptobrachium boringii*, *Megophrys sangzhiensis*, *Polypedates megacephalus, Pelophylax nigromaculatus*, *Rhacophorus dennysi*, *Rana omeimontis*	VIPFVASVAAEMMHHVYCAASKRCKN	26	2	43.2	2863.42
PeNi8	*Bufo gargarizans*, *Megophrys sangzhiensis*, *Polypedates megacephalus*, *Pelophylax nigromaculatus*, *Rhacophorus dennysi*, *Rana omeimontis*	GILLNTLKGAAKNVAGVLLDKLKCKITGGC	30	4	63.0	3012.69
PeNi9	*Leptobrachium boringii, Megophrys sangzhiensis*, *Polypedates megacephalus*, *Pelophylax nigromaculatus*, *Rhacophorus dennysi*, *Rana omeimontis*	GLLGKILGVGKKVLCGVSGLC	21	3	62.2	2014.55
PeNi10	*Leptobrachium boringii*, *Polypedates megacephalus*, *Pelophylax nigromaculatus*, *Rhacophorus dennysi*, *Rana omeimontis*	GLLLDTVKGAAKNVAGILLNKLKCKVTGDC	30	3	61.8	3056.70
PeNi11	*Leptobrachium boringii*, *Polypedates megacephalus*, *Pelophylax nigromaculatus*, *Rhacophorus dennysi*, *Rana omeimontis*	GILTDTLKGAAKNVAGVLLDKLKCKITGGC	30	3	61.8	3001.62
PeNi14	*Bufo gargarizans*, *Polypedates megacephalus*, *Pelophylax nigromaculatus*, *Rana omeimontis*	GLWTTIKEGVKNFSVGVLDKIRCKITGGC	29	3	67.00	3123.71
PoSn1	*Polistes snelleni*	ISIKEALEHSFFHTVPRKWCKKH	23	3	30.4	2822.31
PoSn2	*Polistes snelleni*	TALKSLSILKKLAKLNM	17	4	23.7	1872.37
RaCa15	*Rana catesbeiana*	FLPVVAGLAAKVLPSIICAVTKKC	24	3	67.0	2442.09
RaOm2	*Rana omeimontis*	GILSGLLGAGKKIVCGLSGMC	21	2	80.0	1977.47
RaOm3	*Rana omeimontis*	GIFSLIKGAAKVVAKGLGK	19	4	67.0	1857.30
RaOm4	*Rana omeimontis*	GLLGKVLGVGKKVLCGVSGRC	21	4	67.0	2043.55
RaSi1	*Allobates femoralis*, *Pristimantis toftae*, *Ranitomeya sirensis*	GLVGKLVKGGLKLIGHVANG	20	3	36.9	1930.35
RaSy2	*Rana sylvatica*	EEQRFLPVVAGLAAKVLPSIICAVTKKC	28	2	21.9	2984.64
TeBi1	*Tetramorium bicarinatum*	KIKIPWGKVKDFLVGGMKAVGKK	23	6	45.00	2528.17
TeRu2	*Temnothorax rugatulus*	AFVRILCYCCPRRIKRR	17	6	31.9	2153.70
TeRu4	*Temnothorax rugatulus*	SWLSKSVKKLVNKKNYTRLEKLAKKKLFNE	30	8	25.5	3622.33

* Net charge at pH = 7.

## Data Availability

Characterization results of AMPs with respect to known AMPs, MIC ranges of all tested peptides and selectivity index ranges of bioactive AMPs against the tested bacteria can be found in the Appendix A.

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
