# Peer review of "Associating Biological Activity and Predicted Structure of Antimicrobial Peptides from Amphibians and Insects"

_antibiotics, 2022, doi:10.3390/antibiotics11121710_

Round 1
Reviewer 1 Report
Amelia Richter et al extracted the AMPs from amphibians and insects,predicted and explained the action against few bacterial species.
1. This article is very interesting and done extraordinary work to identify novel anti bacterial peptide to treat.
2. In what basis the author only choose to test these three bacterial or the author done any predictions to choose the tested bacteria.
3. It would be more interest for readers if the author could superimpose the predicted 3D structure based on the family sequence.
4. Most of the AMPs having repeating motifs in each family. Does the repeating aminoacid motifs have any significant action or essential towards anti micorbial activity?.
Author Response
Point 1: This article is very interesting and done extraordinary work to identify novel anti bacterial peptide to treat.
Response 1: Thank you for your interest and support.
Point 2: In what basis the author only choose to test these three bacterial or the author done any predictions to choose the tested bacteria.
Response 2: We chose these 3 bacterial species as they encompass representative Gram-negative and Gram-positive species from the WHO critical pathogen list, as explained on lines 93-94.
Point 3: It would be more interest for readers if the author could superimpose the predicted 3D structure based on the family sequence.
Response 3: Thanks for the great suggestion. We have included a figure to the supplementary material (Fig. S1), depicting the superimposed structures of peptides in our set for the four AMP families.
Point 4: Most of the AMPs having repeating motifs in each family. Does the repeating aminoacid motifs have any significant action or essential towards anti micorbial activity?.
Response 4: This is a good question and an intriguing direction of research. We did not analyze the effect of individual amino acids here, only their structure. This is an open question that warrants additional scrutiny.
Reviewer 2 Report
I thank the authors for this interesting study. This study presents the relationship between the predicted AMPs (antimicrobial peptides) structure and observed antibacterial activity. The relationship between AMPS and observed broad spectrum antibacterial activity against gram-negative and gram-positive bacteria is discussed. It is a promising study which shows the detailed correlation between the activity of the AMPs and the in silico structure characterization. The study design is robust, and the results are advocating the hypothesis made for the study. However, there a few concerns as follows:
- Figure 1 The bioactivity of each peptide is shown against three organisms tested as high, moderate and low. Please include the bioactivity quantitative data or range which is used to select the criteria for each peptide as supplementary.
- Figure 2A. It is a good attempt to sow data all in one graph to have deep insight toward the antibacterial activity of each peptide. However, it is difficult to draw a conclusion by merely looking at the graph, as most of the points overlap on a tight scale. Can you please choose another form of graphical representation to show the data or add a table by the side of the graph showing the values.
- Results 2.3, lines156 – 158. Please include the quantitative data in terms of MIC or the range for the selected active peptides.
- For hemolytic assay, porcine blood is used. What was the rationale for using porcine blood instead of human blood, please include.
- Line 195, 197, “more variation in cell death..” please use cytotoxicity instead of cell death as cell death is not evaluated. Please change the word lethal in line 198 to toxic as it is not the lethal dose which is evaluated. Similarly, change the term cell death to cytotoxicity in Methods, section 4.6 and include reference.
- Methods-section 4.2. Please include if each type of bacterial isolates were grown in MHB to make isolates. Also mention if NaCl, rabbit serum, horse blood or any other thing was added in growing any of the bacterial isolates.
- Is there a specific reason to use MHB for all three types of bacteria, please include with reference.
- Discussion, Lines 294-296, 298-299. Table S2 shows the therapeutic index. But the present study shows in vitro data not in vivo data. None of the experiments were performed in in vivo system. The therapeutic index is a ratio that compares the blood concentration at which a drug causes a therapeutic effect to the amount that causes death (in animal studies) or toxicity (BG Katzung, AJ Trevor - 2012).
- Please include Selectivity index data instead of therapeutic index which is the ratio of CC50/IC50 with appropriate reference.
- Include conclusions section before methods.
Author Response
Point 1: Figure 1 The bioactivity of each peptide is shown against three organisms tested as high, moderate and low. Please include the bioactivity quantitative data or range which is used to select the criteria for each peptide as supplementary.
Response 1: Thank you for your guidance and interest in our work. We have included a description of the MIC range that we used to define the high, moderate and low categories in the figure caption. Additionally, we have added a table to the supplementary material (new table S1) with quantitative data for bioactivity for each peptide.
Point 2: Figure 2A. It is a good attempt to sow data all in one graph to have deep insight toward the antibacterial activity of each peptide. However, it is difficult to draw a conclusion by merely looking at the graph, as most of the points overlap on a tight scale. Can you please choose another form of graphical representation to show the data or add a table by the side of the graph showing the values.
Response 2: Thank you for your feedback. As indicated in our response to the above comment, we added a supplementary table (S1) showing the precise experimental data for each peptide.
Point 3: Results 2.3, lines156 – 158. Please include the quantitative data in terms of MIC or the range for the selected active peptides.
Response 3: We have included a definition of active peptides at the corresponding lines in the main manuscript text and a comment directing readers to supplementary table S1.
Point 4: For hemolytic assay, porcine blood is used. What was the rationale for using porcine blood instead of human blood, please include.
Response 4: Porcine blood is a cost-effective and generalizable model for high-throughput toxicity screening of peptides in mammalian red blood cells. This rationale is now mentioned on line 164.
Point 5: Line 195, 197, “more variation in cell death..” please use cytotoxicity instead of cell death as cell death is not evaluated. Please change the word lethal in line 198 to toxic as it is not the lethal dose which is evaluated. Similarly, change the term cell death to cytotoxicity in Methods, section 4.6 and include reference.
Response 5: Good catch. Thank you. We have changed the use of cell death to refer to cell viability instead throughout the manuscript.
Point 6: Methods-section 4.2. Please include if each type of bacterial isolates were grown in MHB to make isolates. Also mention if NaCl, rabbit serum, horse blood or any other thing was added in growing any of the bacterial isolates.
Response 6: The use of MHB in preparation of isolates is included in line 426. A sentence was added to clarify that no additives were used in growing the bacterial isolates mentioned herein.
Point 7: Is there a specific reason to use MHB for all three types of bacteria, please include with reference.
Response 7: The protocol for antimicrobial susceptibility testing has been standardized based on CLSI guidelines, which suggests the use of MHB. CLSI guidelines were cited at line 433.
Point 8: Discussion, Lines 294-296, 298-299. Table S2 shows the therapeutic index. But the present study shows in vitro data not in vivo data. None of the experiments were performed in in vivo system. The therapeutic index is a ratio that compares the blood concentration at which a drug causes a therapeutic effect to the amount that causes death (in animal studies) or toxicity (BG Katzung, AJ Trevor - 2012).
Response 8: Thank you for your comment. We have adjusted the use of the term ‘therapeutic index’ to ‘selectivity index’ to improve clarity.
Point 9: Please include Selectivity index data instead of therapeutic index which is the ratio of CC50/IC50 with appropriate reference.
Response 9: Changed, as suggested.
Point 10: Include conclusions section before methods.
Response 10: Changed, as suggested.
Reviewer 3 Report
The manuscript “Associating biological activity and predicted structure of anti-microbial peptides from amphibians and insects”, studies a set of 88 AMPs isolated from amphibians and insects that demonstrate activity against a panel of 3 bacterial pathogens. The AMPs are found to have undetectable or low toxicity to human cell lines and red blood cells. Antimicrobial activity is then associated with helical secondary structures adopted by the peptides, as determined by several AI-based software programs including Colabfold. The authors thereby provide a categorization of AMPs on the basis of structure (helical content) and antimicrobial activity.
Overall, I found the organization of the manuscript to be excellent. The Introductory material concisely and sufficiently reviews the current state of AMP research. The authors make a good point that the majority of AMPs are of unknown structure (due to the cost and labor involved with structural elucidation methods and the sheer number of AMPs known), therefore substantiating the importance of this work. The figures used in this work clearly demonstrate the associated findings of the experiments between AI-determined AMP structures and antimicrobial activity. This demonstrates a viable and economical strategy for finding structure-activity relationships for further work in the field.
I have no suggestions for improvement and recommend this manuscript for publication.
Author Response
Point 1: I have no suggestions for improvement and recommend this manuscript for publication.
Response 1: Thank you for your support.